# Comparative Evaluation of Tomato Hybrids and Inbred Lines for Fruit Quality Traits

**Ilias D. Avdikos [1,*], Rafail Tagiakas [1], Pavlos Tsouvaltzis [2], Ioannis Mylonas [3], Ioannis N. Xynias [4] and Athanasios G. Mavromatis [1]**

1. Laboratory of Genetics and Plant Breeding, School of Agriculture, Aristotle University of Thessaloniki, 54124 Thessaloniki, Greece; rafail.tagiakas@gmail.com (R.T.); amavromat@agro.auth.gr (A.G.M.)
2. Laboratory of Vegetable Crops, School of Agriculture, Aristotle University of Thessaloniki, 54124 Thessaloniki, Greece; ptsouv@agro.auth.gr
3. Institute of Plant Breeding and Genetic Resources, Hellenic Agricultural Organization–"Demeter", 57001 Thessaloniki, Greece; ioanmylonas@yahoo.com
4. School of Agricultural Sciences, University of Western Macedonia, Terma Kontopoulou, 53100 Florina, Greece; ioannis_xynias@hotmail.com
* Correspondence: avdikose@agro.auth.gr; Tel.: +30-6973216350

**Abstract:** Tomato is one of the most consumed fruit vegetables globally and is a high dietary source of minerals, fiber, carotenoids, and vitamin C. The tomato is also well known for its nutraceutical chemical content which strengthens human immune systems and is protective against infectious and degenerative diseases. For this reason, there has been recent emphasis on breeding new tomato cultivars with nutraceutical value. Most of the modern tomato cultivars are $F_1$ hybrids, and many of the characteristics associated with fruit quality have additive gene action; so, in theory, inbred vigor could reach hybrid vigor. A sum of 20 recombinant lines was released from the commercial single-cross hybrids Iron, Sahara, Formula, and Elpida, through a breeding process. Those recombinant lines were evaluated during spring–summer 2015 under organic farming conditions in a randomized complete block design (RCBD) experimental design with three replications. A sum of eleven qualitative characteristics of the fruit was recorded on an individual plant basis. Results from this study indicated that the simultaneous selection of individual tomato plants, both in terms of their high yield and desired fruit quality characteristics, can lead to highly productive recombinant lines with integrated quality characteristics. So, inbred vigor can reach and even surpass hybrid vigor. The response to selection for all characteristics evaluated shows additive gene action of all characteristics measured. These recombinant lines can fulfill this role as alternatives to hybrid cultivars and those that possess high nutritional values to function as functional-protective food.

**Keywords:** antioxidants; inbred vigor; fruit quality; heterosis tomato; hybrid vigor; lycopene; recombinant lines

## 1. Introduction

Tomato (*Solanum lycopersicum* L.) is one of the most economically important vegetable crops, being widely grown and appreciated for its organoleptic and nutritional properties. Its cultivation covers 4.76 million hectares with an annual production of 182 million tons [1]. The tomato is one of the most consumed vegetables globally and is a high dietary source of minerals (potassium, magnesium, phosphorus), fibers, carotenoids, and vitamin C [2]. Many nutraceutical compounds produced by tomato are protective against infectious [3,4] and degenerative human diseases, such as cardiovascular diseases [5] or certain cancers [6,7] and strengthen our immune system [8]. For this reason, there has been recent emphasis on breeding new tomato cultivars with high nutraceutical value.

The cultivated tomato is a self-pollinated species, and its mating system determines the choice of the suitable breeding method for the improvement of quality traits. In the past,

tomato cultivars were inbred lines produced through the selection process of populations after several selfing generations [9]. Nevertheless, most of the modern cultivars are $F_1$ hybrids, produced by crossing two complementary inbred lines. $F_1$ hybrids are utilized since they drive benefit from heterosis for several traits and protect breeders' innovations. Generally, modern tomato hybrids are uniform and highly productive. They originate from a reduced gene pool, and are bred for a few traits of major interest for the fresh market or and processing industries. However, this often results in the absence of those fruit quality traits important to consumers, such as flavor [10].

Heterosis or hybrid vigor is a widely documented phenomenon in tomato and is manifested as an improved performance of $F_1$ hybrids over both parents. More than 50%–60% of tomato heterotic performance studies focused on heterosis for yield and yield components [11]. This percentage was approximately stable even though tomato breeders' efforts have strongly focused on nutritional value, safety and sensory quality of a food product, tolerance to abiotic stress, etc. during the last two decades. Based on these reports and numerous other studies, it seems that heterosis in tomato was observed for many quantitative traits, almost all of them being of breeding interest. Heterosis has been found for characteristics related to fruit quality, for instance, pericarp thickness [12–15], total soluble solids, and dry matter content [12–14,16–20], lycopene and β-carotene content in fruits [21–24], and number of locules per fruit [25].

As mentioned previously, a significant number of tomato genetic studies and breeding programs in recent years emphasized improving fruit nutritional value, and sensory and market quality. This entailed determining the genetic basis of several characteristics related to these traits. Lycopene content was found to have dominant gene action [13], total soluble solids additive [26], β-carotene content additive and non-additive [13,27], pericarp thickness and dry matter content additive [28–31], number of locules per fruit non-additive and additive [32–34], fruit shape additive and non-additive [34,35], and fruit width and length additive [36].

In self-fertilized crops like tomato, where additive genetic variation predominates, it is always feasible to fix and transgress heterosis. The most significant advantage from emphasizing selection for inbred instead of hybrid vigor is the constant and efficient exploitation of additive genetic variation responsible for genetic advances through selection. Fasoula and Fasoula [37] pointed out that in such cultivars as tomato, inbred vigor could be superior to hybrid vigor. Thus, this work aims to compare hybrid vigor with inbred vigor in terms of tomato's fruit quality characteristics. Based on previous research, many of the characteristics associated with tomato fruit quality have additive gene action; so in theory, inbred vigor could be similar to hybrid vigor.

## 2. Materials and Methods

### 2.1. Plant Material

The commercial tomato $F_1$ single-cross hybrids 'Iron', 'Sahara', 'Formula', and 'Elpida' were used as the genetic source materials. The above cultivars (cvrs.) cover 20% (cvrs. 'Iron' and 'Sahara') and 50% (cvrs. 'Formula' and 'Elpida') of the total tomato cropping area in Greece [38]. The traditional cultivar 'Makedonia', one of the most popular tomato inbred lines in Greece, was used as control. 'Makedonia' is a cultivar developed at the Agricultural Research Center of Northern Greece (ARCNG), by using the pure line breeding method of a local tomato population. This important indeterminate cultivar has been widely used in the traditional farming and low-input cropping systems. It is characterized by preferable physicochemical and sensory properties with attractive and tasty fresh fruits.

### 2.2. Methodology

Mass selection of the hybrids 'Iron' and 'Sahara' was applied for two generations to produce $F_3$ segregating generations, and this was followed by recurrent selection to produce $HS_5$ recombinant lines. These lines were produced after pollination of selected plants by a mixture of pollen from the previous cycle's selected plants. Pollen donors were

those plants that combined the desired performance in terms of yield components and fruit quality characteristics [39]. Finally, four recombinant half sib tomato lines, two from each of the two parental hybrids were selected.

Two commercial tomato hybrids that resemble 'Sahara' and 'Iron' in $F_2$ generation inbreeding depression were determined from 20 commercial tomato hybrids. 'Formula' and 'Elpida' hybrids were selected for continuing the breeding process. Pedigree selection under low plant density was applied for two seasons to these two hybrids, in a honeycomb design. This was followed by pedigree selection and evaluation of progenies at recommended plant density conditions using a randomized complete block design (RCBD) (Figure 1). At the end of all pedigree processes ($F_5$), a sum of sixteen recombinant lines, eight from each hybrid, were selected based on all the desirable characteristics, including earliness, productivity, and fruit quality traits.

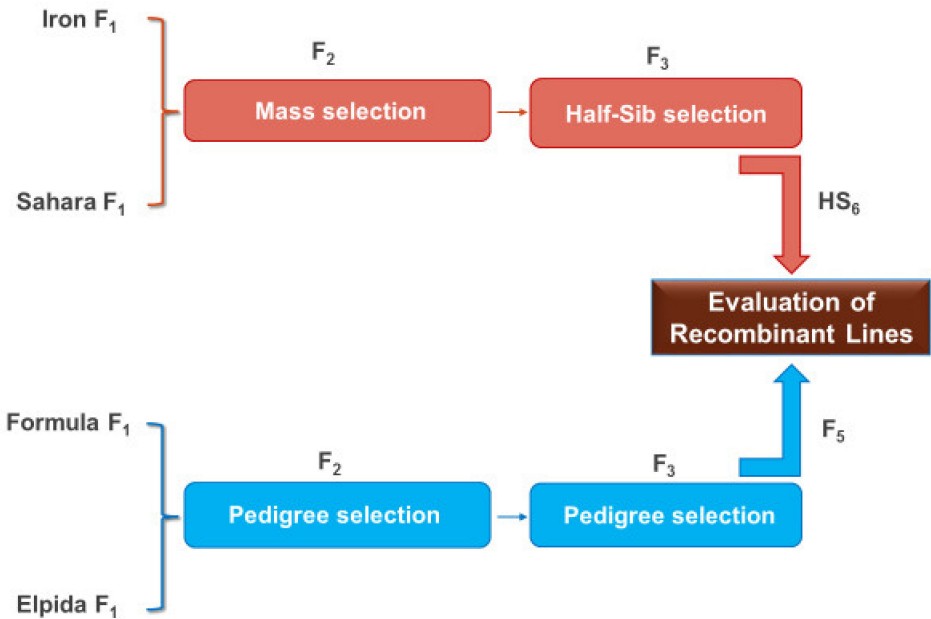

**Figure 1.** Breeding process applied for the production of new recombinant lines.

### 2.3. Selection and Assessment Procedure

A sum of 20 recombinant lines obtained from the commercial single-cross hybrids Iron, Sahara, Formula, and Elpida, were evaluated. The experiment was performed under organic farming conditions, during spring–summer 2015, in a non-heated greenhouse at the farm of the ARCNG, Thermi–Thessaloniki (40°32 N lat., 22°59 E long., 3 m alt.). The experimental plant material was prepared conventionally. Uniform seedlings were hand-transplanted on 15 April 2015, at a density of 2 plants/m$^2$. The high wire one-stem training system was used. The greenhouse was shaded in June, July, and August. A randomized complete block design (RCBD) experimental design was used, with three replications, with each experimental unit consisting of ten plants. Organic (low input) cropping practices were followed (manure, field rotation with legumes, soil mulching using a biodegradable film, and no chemical applications). Composted poultry manure, 3 t ha$^{-1}$ (dry weight) was used as the primary nutrient source. All observations were taken on an individual plant basis, and for each entry, qualitative characteristics of the fruit were determined.

### 2.4. Traits Evaluated

Two red ripe stage fruits were harvested (the first and second fruit of the second cluster of each plant), and their nutritional composition was evaluated in the laboratory of vegetable crops at the farm of Aristotle University of Thessaloniki. The total solids were determined on blended samples of two sections of each fruit after oven drying at 70 °C. The total soluble solids (TSS) were determined with the use of an Atago PR-100

hand refractometer on the juice taken from each sample. The other fruit sections of each sample were frozen. These samples were later thawed and mixed in a blender. The blended material was used for determination of ascorbic acid, total soluble phenols, total carotenoids, and lycopene content, as well as total antioxidant capacity.

For the extraction of ascorbic acid, 25 g of blended tomato fruit material was homogenized with 50 mL of 1% oxalic acid solution and filtered through Whatman No 1 filter paper. Ascorbic acid content was measured in the filtrate using a RQflex reflectrometer (Merck, Darmstadt, Germany).

For the determination of total soluble phenols and total antioxidant capacity, 5 g of the blended material was homogenized with 25 mL of 80% methanol, centrifuged at $5000\times g$ for 20 min and filtered through Whatman No 1 filter paper. Total soluble phenols were determined in the supernatant following the method of Scalbert et al. [40], using gallic acid for the standard curve. The results are expressed as µg gallic acid equivalents (GAE) per g fresh weight.

Total antioxidant capacity was determined following the method of Brand-Willams et al. [41]. In particular, 200 µL of the supernatant was added to 2800 µL 100 µM 2,2-diphenyl-1-picrylhydrazyl (DPPH) solution in 95% methanol in a test tube, vortexed and kept in darkness for 1 h at room temperature. The absorbance was measured at 517 nm in a spectrophotometer and ascorbic acid was used for the standard curve. The radical scavenging capacity of DPPH, representing the total antioxidant capacity, was expressed as mg ascorbic acid equivalents (AEAC) per 100 g fresh weight.

The extraction and the determination of total carotenoids, β-carotene and lycopene content were performed according to Lichtenthaler and Wellburn [42] and D'Souza et al. [43]. For the extraction of carotenoids, 1 g of blended material was mixed with 15 mL of 80% acetone, vortexed, centrifuged at 14,000 rpm for 10 min at 20 °C and the supernatant filtered through Whatman No 1 in 25 mL volumetric vials. Another 10 mL of 80% acetone was added to the residue, and samples were shaken at 150 rpm for 10 min. The samples were filtered again and added to the previous filtrates. The vials were filled with 80% acetone and the absorbance of the filtrates was measured in a spectrophotometer at 470, 662, and 645 nm for the determination of total carotenoids and at 450 and 503 nm for β-carotene and lycopene. Total and individual carotenoid contents are expressed as µg/g f.w.

The thickness of fruit pericarp (mm), the size of the peduncle inside the fruit (mm), the number of fruit locules, and the intensity of the internal color of fruit at maturity (1:weak, 5:strong) were also determined.

### 2.5. Statistical Analyses

A randomized complete block design (RCBD) with three replications was used. Data were subjected to analysis of variance (ANOVA), with genotypes as fixed factor and blocks as random, and Scott–Knott cluster test at 5% probability and significance level [44]. The correlations between the characteristics were performed using the Pearson correlation coefficient (rp) and were determined for significance level a = 0.01. All statistical analyses were performed using the SPSS software package (ver. 18, SPSS Inc., Chicago, IL, USA).

## 3. Results

Five recombinant lines originating from the 'Formula' hybrid had an inbred vigor up to 25% for TSS content. 'Formula' $F_5$-1 recombinant line also outperformed the parental hybrid (Table 1). The recombinant lines of 'Elpida', had lower TSS content compared to the original hybrid by 9% to 25%. Both 'Iron' and 'Sahara' recombinant lines did not differ from their respective hybrids in TSS. 'Iron' $HS_6$-3 and 'Sahara' $HS_6$-2 lines showed a small amount of inbred vigor (2% and 3%, respectively) compared to their hybrids. The domestic inbred cultivar 'Makedonia' showed one of the highest TSS values in this characteristic with 6.37 °Brix. However, 'Formula' $F_5$-1 reached 6.50 °Brix, having the best performance of all entries (Table 1).

**Table 1.** Average fruit values for total soluble solids (°Brix) and total solids (%), and inbred vigor/depression (% of corresponding hybrid), for tomato hybrids and recombinant inbred lines [45].

| Entry | Total Soluble Solids (°Brix) | | Total Solids (%) | |
|---|---|---|---|---|
| | $\bar{x}$ | Inb. Vig/Dep (%) | $\bar{x}$ | Inb. Vig/Dep (%) |
| Formula $F_1$ | 5.20 c * | 100 | 5.68 c | 100 |
| Formula $F_5$-1 | 6.50 a | 125 | 6.22 b | 110 |
| Formula $F_5$-2 | 4.77 d | 92 | 5.84 b | 103 |
| Formula $F_5$-3 | 5.50 c | 106 | 6.17 b | 109 |
| Formula $F_5$-4 | 4.77 d | 92 | 5.37 c | 95 |
| Formula $F_5$-5 | 4.87 d | 94 | 5.58 c | 98 |
| Formula $F_5$-6 | 5.48 c | 105 | 6.45 a | 114 |
| Formula $F_5$-7 | 5.67 c | 109 | 6.66 a | 117 |
| Formula $F_5$-8 | 5.38 c | 103 | 6.34 a | 112 |
| Elpida $F_1$ | 5.92 b | 100 | 6.72 a | 100 |
| Elpida $F_5$-1 | 5.07 d | 86 | 5.83 b | 87 |
| Elpida $F_5$-2 | 4.97 d | 84 | 5.95 b | 89 |
| Elpida $F_5$-3 | 4.42 e | 75 | 5.34 c | 79 |
| Elpida $F_5$-4 | 4.64 d | 78 | 5.84 b | 87 |
| Elpida $F_5$-5 | 5.27 c | 89 | 5.95 b | 89 |
| Elpida $F_5$-6 | 5.40 c | 91 | 5.98 b | 89 |
| Elpida $F_5$-7 | 5.24 c | 89 | 5.94 b | 88 |
| Elpida $F_5$-8 | 5.38 c | 91 | 5.99 b | 89 |
| Iron $F_1$ | 5.63 c | 100 | 6.21 b | 100 |
| Iron $HS_6$-2 | 5.47 c | 97 | 6.05 b | 97 |
| Iron $HS_6$-3 | 5.77 c | 102 | 6.56 a | 106 |
| Sahara $F_1$ | 4.31 e | 100 | 5.21 c | 100 |
| Sahara $HS_6$-1 | 3.90 e | 90 | 4.58 d | 88 |
| Sahara $HS_6$-2 | 4.45 e | 103 | 6.04 b | 116 |
| Makedonia | 6.37 a | — | 6.96 a | — |

* Entries with the same letter within a column indicate no significant difference, according to Scott–Knott cluster test ($\alpha = 0.05$).

All recombinant inbred lines derived from 'Formula' hybrid had similar or better total solids content than the original F1 hybrid. Three lines, 'Formula' $F_5$-6, 'Formula' $F_5$-7, and 'Formula' $F_5$-8 had total solid contents that indicated inbred vigor of 14%, 17%, and 12%, respectively. In comparison, recombinant lines originating from 'Elpida' hybrid had an inbreeding depression between 11% to 21% for total solids, while recombinant lines 'Iron' $HS_6$-3 and 'Sahara' $HS_6$-2, exceeded the parental hybrid by 6% and 16%, respectively. The domestic cultivar 'Makedonia' had the highest total solids values, although there were four recombinant lines along with 'Elpida' hybrid, that did not differ from this cultivar.

All tomato genetic material evaluated did not differ for β-carotene concentration in fruit. Recombinant lines, 'Formula' $F_5$-3, 'Formula' $F_5$-6, 'Formula' $F_5$-7, and 'Sahara' $HS_6$-2 exceeded their parental hybrids by 2%, 15%, 11%, and 1%, respectively (Table 2). The domestic cultivar 'Makedonia' had the best performance with 15.57 μg/g f.w. In addition, recombinant lines 'Elpida' $F_1$ hybrid, 'Elpida' $F_5$-6, and 'Elpida' $F_5$-5, also revealed high β-carotene contents of 12.45, 12.37, and 12.35 μg/g f.w., respectively.

For fruit carotenoid content, the recombinant lines 'Formula' $F_5$-7 and 'Formula' $F_5$-8 had 39.13 and 39.04 μg/g f.w., respectively), which was similar to the parental hybrid (Table 2). Similarly, three recombinant 'Elpida' lines, 'Elpida' $F_5$-3, 'Elpida' $F_5$-5, and 'Elpida' $F_5$-6, did not differ ($p < 0.05$) from 'Elpida' $F_1$ hybrid for total carotenoids. The two recombinant lines of 'Iron' did not differ from the hybrid with 'Iron' $HS_6$-2, having a small inbred vigor of 3%. In 'Sahara' and its recombinant lines, 'Sahara' $HS_6$-2 possessed a non-significant inbred vigor of 13%. 'Formula' and 'Elpida' hybrids as well as recombinant lines, 'Elpida' $F_5$-3, 'Elpida' $F_5$-5, and 'Sahara' $HS_6$-2, had higher carotenoid contents than the domestic cultivar 'Makedonia' (41.75 μg/g f.w.) in fruit carotenoids (Table 2).

**Table 2.** Average fruit β-carotene (μg/g f.w.), total carotenoids (μg/g f.w.), and inbred vigor/depression (% of corresponding hybrid), for tomato hybrids and recombinant inbred lines.

| Entry | β-Carotene (μg/g f.w.) | | Carotenoids (μg/g f.w.) | |
|---|---|---|---|---|
| | $\bar{x}$ | Inb. Vig/Dep (%) | $\bar{x}$ | Inb. Vig/Dep (%) |
| Formula $F_1$ | 10.18 a * | 100 | 43.44 a | 100 |
| Formula $F_5$-1 | 10.04 a | 99 | 37.35 b | 86 |
| Formula $F_5$-2 | 8.21 a | 81 | 27.34 b | 63 |
| Formula $F_5$-3 | 10.42 a | 102 | 33.16 b | 76 |
| Formula $F_5$-4 | 7.76 a | 76 | 33.73 b | 78 |
| Formula $F_5$-5 | 8.19 a | 80 | 33.08 b | 76 |
| Formula $F_5$-6 | 11.73 a | 115 | 35.52 b | 82 |
| Formula $F_5$-7 | 11.27 a | 111 | 39.13 a | 90 |
| Formula $F_5$-8 | 9.62 a | 94 | 39.04 a | 90 |
| Elpida $F_1$ | 12.45 a | 100 | 47.51 a | 100 |
| Elpida $F_5$-1 | 8.04 a | 65 | 34.33 b | 72 |
| Elpida $F_5$-2 | 9.31 a | 75 | 33.52 b | 71 |
| Elpida $F_5$-3 | 10.13 a | 81 | 45.21 a | 95 |
| Elpida $F_5$-4 | 10.26 a | 82 | 33.15 b | 70 |
| Elpida $F_5$-5 | 12.35 a | 99 | 43.31 a | 91 |
| Elpida $F_5$-6 | 12.37 a | 99 | 41.26 a | 87 |
| Elpida $F_5$-7 | 10.59 a | 85 | 33.53 b | 71 |
| Elpida $F_5$-8 | 9.06 a | 73 | 27.86 b | 59 |
| Iron $F_1$ | 10.37 a | 100 | 34.43 b | 100 |
| Iron $HS_6$-2 | 10.39 a | 100 | 35.49 b | 103 |
| Iron $HS_6$-3 | 8.88 a | 86 | 30.06 b | 87 |
| Sahara $F_1$ | 11.11 a | 100 | 40.78 a | 100 |
| Sahara $HS_6$-1 | 10.94 a | 98 | 37.55 b | 92 |
| Sahara $HS_6$-2 | 11.21 a | 101 | 46.01 a | 113 |
| Makedonia | 15.57 a | — | 41.75 a | — |

* Entries with the same letter within a column indicate no significant difference, according to Scott–Knott cluster test (α = 0.05).

All tomato hybrids and recombinant lines did not differ for fruit lycopene content. However, the recombinant lines of both 'Formula' and 'Elpida' had inbreeding depressions ranging from 4% to 45% compared to the parental hybrids. 'Iron' $HS_6$-2 and 'Sahara' $HS_6$-2 showed an inbred vigor of 6% and 21%, respectively, compared to their parental hybrids. The domestic cultivar 'Makedonia' had one of the lowest lycopene contents of 17.91 μg/g f.w., while recombinant line 'Sahara' $HS_6$-2 had the highest of 27.44 μg/g f.w.

For fruit phenols content, all recombinant 'Formula' lines showed an increased inbred vigor of 7% to 73% (Table 3). The recombinant line 'Elpida' $F_5$-4 was similar to the parental hybrid, while 'Elpida' $F_5$-1 and 'Elpida' $F_5$-6 exceeded the hybrids' values by 5% to 10%. In 'Iron', the recombinant lines exhibited a small inbreeding depression from 5%-19%, while 'Sahara' $HS_6$-2 showed an inbred vigor increase of 5%. The domestic variety 'Makedonia' had the greatest fruit phenols content at 0.29 μg/g f.w.

Regarding tomato fruit total antioxidant content, four recombinant lines of 'Formula' showed an inbred vigor ranging from 4% to 37% (Table 3). The 'Formula' $F_5$-1 line had 27.44 mg Asc/100 g f.w., which was greater than the parental hybrid and all other recombinant lines. Although six recombinant lines of 'Elpida' had vigor increases up to 15%, all were similar to the parental hybrid. Compared with the parental hybrid the recombinant line 'Iron' $HS_6$-2 had a 10% inbreeding depression while line 'Iron' $HS_6$-3 had similar inbred vigor. For 'Sahara' recombinant lines, 'Sahara' $HS_6$-1 had a slight inbreeding depression of 18%, while 'Sahara' $HS_6$-2 had a 9% inbred vigor. The domestic variety 'Makedonia' had the highest antioxidant content with 28.13 mg Asc/100 g f.w.

**Table 3.** Average fruit lycopene (µg/g f.w.), phenols (µg/g f.w.), antioxidants (µg Asc/100g f.w.) and inbred vigor/depression (% of corresponding hybrid), for tomato hybrids and recombinant inbred lines.

| Entry | Lycopene (µg/g f.w.) | | Phenols (µg/g f.w.) | | Antioxidants (mg Asc/100 g f.w.) | |
|---|---|---|---|---|---|---|
| | $\bar{x}$ | Inb. Vig/Dep (%) | $\bar{x}$ | Inb. Vig/Dep (%) | $\bar{x}$ | Inb. Vig/Dep (%) |
| Formula $F_1$ | 26.63 a * | 100 | 0.153 p | 100 | 20.06 b | 100 |
| Formula $F_5$-1 | 22.11 a | 83 | 0.257 b | 173 | 27.44 a | 137 |
| Formula $F_5$-2 | 15.12 a | 57 | 0.213 d | 140 | 20.77 b | 104 |
| Formula $F_5$-3 | 19.93 a | 75 | 0.173 m | 113 | 19.09 b | 95 |
| Formula $F_5$-4 | 19.16 a | 72 | 0.160 o | 107 | 21.01 b | 105 |
| Formula $F_5$-5 | 18.61 a | 70 | 0.180 k | 120 | 19.87 b | 99 |
| Formula $F_5$-6 | 20.48 a | 77 | 0.173 m | 113 | 17.98 b | 90 |
| Formula $F_5$-7 | 22.81 a | 86 | 0.197 g | 133 | 17.00 b | 85 |
| Formula $F_5$-8 | 21.88 a | 82 | 0.193 h | 127 | 21.44 b | 107 |
| Elpida $F_1$ | 27.25 a | 100 | 0.197 g | 100 | 19.51 b | 100 |
| Elpida $F_5$-1 | 19.31 a | 71 | 0.207 e | 105 | 21.59 b | 111 |
| Elpida $F_5$-2 | 18.51 a | 68 | 0.177 l | 90 | 19.54 b | 100 |
| Elpida $F_5$-3 | 26.21 a | 96 | 0.167 n | 85 | 18.69 b | 96 |
| Elpida $F_5$-4 | 18.15 a | 67 | 0.203 f | 100 | 22.11 b | 113 |
| Elpida $F_5$-5 | 26.00 a | 95 | 0.183 j | 90 | 19.90 b | 102 |
| Elpida $F_5$-6 | 23.00 a | 84 | 0.223 c | 110 | 19.31 b | 99 |
| Elpida $F_5$-7 | 18.18 a | 67 | 0.177 l | 90 | 22.50 b | 115 |
| Elpida $F_5$-8 | 15.04 a | 55 | 0.187 i | 95 | 22.16 b | 114 |
| Iron $F_1$ | 19.26 a | 100 | 0.207 e | 100 | 22.07 b | 100 |
| Iron $HS_6$-2 | 20.43 a | 106 | 0.173 m | 81 | 19.83 b | 90 |
| Iron $HS_6$-3 | 17.29 a | 90 | 0.200 g | 95 | 22.01 b | 100 |
| Sahara $F_1$ | 22.62 a | 100 | 0.193 h | 100 | 23.34 b | 100 |
| Sahara $HS_6$-1 | 21.08 a | 93 | 0.183 j | 95 | 19.22 b | 82 |
| Sahara $HS_6$-2 | 27.44 a | 121 | 0.203 f | 105 | 25.49 a | 109 |
| Makedonia | 17.91 a | — | 0.290 a | — | 28.13 a | — |

* Entries with the same letter within a column indicate no significant difference, according to Scott–Knott cluster test ($\alpha = 0.05$).

For pericarp thickness, the lines originating from 'Formula' did not differ. However, the recombinant line 'Elpida' $F_5$-4 had higher pericarp thickness (10.27 mm) than all other genetic 'Elpida' materials, including the parental hybrid. In comparison, the recombinant line 'Elpida' $F_5$-3 had a thinner pericarp than other 'Elpida' genetic materials, at 7.28 mm. The recombinant lines derived from 'Iron' and 'Sahara' hybrids did not differ in pericarp thickness from their original hybrids. In addition, 'Makedonia' had the lowest values for this characteristic.

For the undesirable feature of peduncle size inside the fruit, most recombinant 'Formula' lines had higher values than the hybrid, while the recombinant lines of 'Elpida' and 'Iron' tended to have lower values (Table 4). The recombinant line 'Sahara' $HS_6$-2 exhibited a smaller peduncle size inside the fruit (1.43 cm) than the parental hybrid (1.53 cm) (Table 4).

For the number of fruit locules, most recombinant lines exhibited inbreeding depression (Table 5). However, three lines were exceptions having increased vigor from their parental hybrids, 'Formula' $F_5$-3, 'Elpida' $F_5$-3, and 'Sahara' $HS_6$-1, at 3%, 8%, and 14%, respectively.

For the intensity of the internal red color in fruit at maturity, most of the recombinant lines of 'Formula' showed vigor improvement, which reached 84% in line 'Formula' $F_5$-8. The recombinant lines 'Elpida' $F_5$-3 and 'Elpida' $F_5$-7 had a slight improvement in inbred vigor at 11% and 4%, respectively, compared to the parental hybrid. The two recombinant 'Iron' lines exhibited slight inbreeding depression of 14% and 19%. Finally, the recombinant line 'Sahara' $HS_6$-2 had a slight inbred vigor increase of 10% for internal fruit color.

**Table 4.** Average fruit pericarp thickness, size of peduncle inside the fruit (mm), and inbred vigor/depression (% of corresponding hybrid), for tomato hybrids and recombinant inbred lines.

| Entry | Pericarp Thickness (mm) | | Peduncle Size inside the Fruit (mm) | |
|---|---|---|---|---|
| | $\bar{x}$ | Inb. Vig/Dep (%) | $\bar{x}$ | Inb. Vig/Dep (%) |
| Formula $F_1$ | 7.20 c * | 100 | 1.46 b | 100 |
| Formula $F_5$-1 | 7.00 c | 97 | 1.48 b | 102 |
| Formula $F_5$-2 | 7.42 c | 103 | 1.66 a | 113 |
| Formula $F_5$-3 | 7.50 c | 104 | 1.62 a | 111 |
| Formula $F_5$-4 | 7.17 c | 100 | 1.95 a | 133 |
| Formula $F_5$-5 | 6.56 c | 91 | 1.85 a | 127 |
| Formula $F_5$-6 | 6.87 c | 95 | 1.49 b | 102 |
| Formula $F_5$-7 | 6.28 c | 87 | 1.28 b | 88 |
| Formula $F_5$-8 | 6.50 c | 90 | 0.97 c | 66 |
| Elpida $F_1$ | 8.40 b | 100 | 1.89 a | 100 |
| Elpida $F_5$-1 | 8.03 b | 96 | 1.56 a | 82 |
| Elpida $F_5$-2 | 8.56 b | 102 | 1.62 a | 86 |
| Elpida $F_5$-3 | 7.28 c | 87 | 1.60 a | 85 |
| Elpida $F_5$-4 | 10.27 a | 122 | 1.85 a | 98 |
| Elpida $F_5$-5 | 9.02 b | 107 | 1.72 a | 91 |
| Elpida $F_5$-6 | 8.73 b | 104 | 1.49 b | 79 |
| Elpida $F_5$-7 | 8.33 b | 99 | 1.03 c | 55 |
| Elpida $F_5$-8 | 8.00 b | 95 | 1.08 c | 57 |
| Iron $F_1$ | 8.28 b | 100 | 1.87 a | 100 |
| Iron $HS_6$-2 | 8.00 b | 97 | 1.77 a | 94 |
| Iron $HS_6$-3 | 7.44 c | 90 | 1.38 b | 74 |
| Sahara $F_1$ | 6.79 c | 100 | 1.71 a | 100 |
| Sahara $HS_6$-1 | 6.28 c | 92 | 1.79 a | 104 |
| Sahara $HS_6$-2 | 6.83 c | 101 | 1.43 b | 84 |
| Makedonia | 5.72 c | — | 1.17 c | — |

* Entries with the same letter within a column indicate no significant difference, according to Scott–Knott cluster test ($\alpha = 0.05$).

**Table 5.** Average number of locules in fruit, internal fruit color intensity at maturity, and inbred vigor/depression (% of corresponding hybrid), for tomato hybrids and recombinant inbred lines.

| Entry | Number of Locules | | Internal Colour of Fruit at Maturity (1:Weak, 5:Strong) | |
|---|---|---|---|---|
| | $\bar{x}$ | Inb. Vig/Dep (%) | $\bar{x}$ | Inb. Vig/Dep (%) |
| Formula $F_1$ | 5.40 a * | 100 | 2.03 c | 100 |
| Formula $F_5$-1 | 4.83 a | 90 | 3.00 a | 148 |
| Formula $F_5$-2 | 3.55 b | 66 | 2.28 b | 112 |
| Formula $F_5$-3 | 5.58 a | 103 | 1.54 c | 76 |
| Formula $F_5$-4 | 3.67 b | 68 | 1.75 c | 86 |
| Formula $F_5$-5 | 4.42 b | 82 | 2.60 b | 128 |
| Formula $F_5$-6 | 4.53 b | 84 | 2.60 b | 128 |
| Formula $F_5$-7 | 5.02 a | 93 | 2.63 b | 129 |
| Formula $F_5$-8 | 4.50 b | 83 | 3.75 a | 184 |
| Elpida $F_1$ | 3.92 b | 100 | 3.13 a | 100 |
| Elpida $F_5$-1 | 3.83 b | 98 | 2.97 a | 95 |
| Elpida $F_5$-2 | 3.42 b | 87 | 2.06 c | 66 |
| Elpida $F_5$-3 | 4.22 b | 108 | 3.46 a | 111 |
| Elpida $F_5$-4 | 3.42 b | 87 | 1.77 c | 57 |
| Elpida $F_5$-5 | 3.32 b | 85 | 1.79 a | 57 |
| Elpida $F_5$-6 | 3.83 b | 98 | 2.43 b | 78 |
| Elpida $F_5$-7 | 3.72 b | 95 | 3.26 a | 104 |
| Elpida $F_5$-8 | 3.83 b | 98 | 2.83 a | 91 |

**Table 5.** *Cont.*

| Entry | Number of Locules | | Internal Colour of Fruit at Maturity (1:Weak, 5:Strong) | |
|---|---|---|---|---|
| | $\bar{x}$ | Inb. Vig/Dep (%) | $\bar{x}$ | Inb. Vig/Dep (%) |
| Iron F$_1$ | 4.72 a | 100 | 2.95 a | 100 |
| Iron HS$_6$-2 | 4.40 b | 93 | 2.40 b | 81 |
| Iron HS$_6$-3 | 4.39 b | 93 | 2.54 b | 86 |
| Sahara F$_1$ | 5.17 a | 100 | 3.17 a | 100 |
| Sahara HS$_6$-1 | 5.89 a | 114 | 1.67 c | 53 |
| Sahara HS$_6$-2 | 5.17 a | 100 | 3.50 a | 110 |
| Makedonia | 5.22 a | — | 3.46 a | — |

* Entries with the same letter within a column indicate no significant difference, according to Scott–Knott cluster test ($\alpha = 0.05$).

## 4. Discussion

Tomato is a 'protective food' since it is rich in minerals, vitamins, antioxidants, and organic acids [46,47]. Beside contributing nutritive elements, color, and flavor to the diet, tomato fruit also has many nutraceutical benefits [48]. Moreover, as a valuable source of antioxidants, or chemo-protective compounds, it is often termed a 'functional food' [49]. The antioxidant potential of tomato results from a mixture of antioxidant biomolecules, including lycopene and phenols [50]. Although breeding efforts in tomato started more than 200 years ago [51], it was not until the 1950s that breeding programs released multi-purpose cultivars that fulfilled several needs like productivity, tolerances to abiotic stresses, broad adaptability to different environments, and early fruit maturity. Following market requirements, traditional breeding programs focused their attention on these external features in combinations with higher yields and disease resistance, ignoring other important aspects such as flavor or functional characteristics. Over the last few decades, flavor has been a focus in breeding programs due to consumers' continuous complaints about the loss of traditional organoleptic characteristics in vegetables [52]. Nowadays, consumers are aware of food products' functional characteristics, and more consumers choose foods based on improved health characteristics. The nutrition importance of tomato in human diets indicates a need to develop cultivars rich in vitamins, nutrients, and antioxidants, besides fruit yield [53].

Since the 1980s, the emphasis of new cultivar development has focused on the production of F$_1$ hybrids [54]. Replacement of inbred lines with hybrids has remarkably increased yield. Thus, the phenomenon of heterosis for yield components and quality traits have been extensively studied. A prerequisite for improvement through heterosis breeding is the knowledge of its extent for yield and quality component characters. Many researchers claim that the use of hybrids in tomato is more due to protection of breeders' research investment than the benefits of heterosis per se [55,56]. Avdikos et al. [45] produced elite inbred recombinant lines that surpassed hybrids in productivity.

The total soluble solids content of tomato fruit is a crucial trait, because it affects the final product flavor and consistency, and determines the final yield after processing. Kumar et al. [57] found that the heterobeltiotic effects of total tomato soluble solids were 32.68%. Comparable results have also been reported by Singh et al. [36], Kumari and Sharma [58], and Graca et al. [59]. In this study, several recombinant lines of 'Formula', 'Iron' and 'Sahara' tomato surpassed their corresponding hybrids, with inbred vigor ranging from 2% to 25%. For total solids, a characteristic found to have a high correlation with total soluble solids (r = 0.71), the recombinant lines had similar behavior having inbred vigor up to 17%.

Tomato is consumed year-round and characterized only by moderate nutritional value. However, it is an essential source of carotenoids, which are protective against infection [3,4] and degenerative diseases such as cardiovascular diseases [5] or certain cancers [6,7]. Breeding has targeted tomato fruit color, which is a basic fruit quality objective, demanded

by markets. This trait is conferred by carotenoid pigments (mainly lycopene and carotene) and reflects significant antioxidant properties. Fruit color improvement has indirectly led to improved tomato functional and nutritional value. This situation explains why breeding for improved carotenoid content is far more advanced than any other bioactive compounds. Kumar et al. [60] found heterosis for total carotenoids content of 29.53%. In our study, two recombinant lines, one of 'Iron' and another of 'Sahara', showed inbred vigor of 3% and 13% from their corresponding hybrids for carotenoid content. For β-carotene, four recombinant lines, three originating from 'Formula' and one from 'Sahara' had inbred vigor ranging from 2% to 15%.

Lycopene is responsible for red color in the tomato fruit, and deep, uniformly red-colored tomato fruits are preferred for both processing and table purposes. Moreover, lycopene is an antioxidant characterized by immuno-stimulatory properties and protects cells against oxidative damage, thereby preventing or reducing the risk of several cancers [61] and cardiovascular disease [62]. Many researchers have reported heterosis in terms of lycopene content in tomato fruit, including Garg et al. [63] (heterosis 134%), Kumar et al. [60] (heterosis 26%), and Kumar et al. [64] (heterosis 60%). As shown in this study, two recombinant lines, one for each 'Iron' and 'Sahara', had inbred vigor up to 21% for lycopene content.

Kumar et al. [60] found that the heterosis level for phenolic compounds reached 39%. In this experiment, almost all hybrids produced recombinant lines with high inbred vigor levels that reached up to 73%. Tomato antioxidants include vitamins such as ascorbic acid and tocopherols, phenolic compounds such as flavonoids, and carotenoids such as beta-carotene, a precursor of vitamin A, as well as lycopene, which is responsible for the red color of the fruit [65–69]. Kumar et al. [60] found 10.42% heterosis for total antioxidants in tomato fruit. In this experiment, 12 recombinant lines derived from tomato hybrids exhibited inbred vigor up to 37%.

Pericarp thickness in tomato fruit is desirable because it imparts firmness, making fruits suitable for canning, better storage, and long-distance transportation [70–72]. Heterosis for pericarp thickness has been described by Patil and Patil [12], Daskaloff et al. [13], and Dod and Kale [14], and for fruit firmness by Wang et al. [73], Resende et al. [74], and Atanassova et al. [75]. Kumar et al. [64] found heterosis of 57% for pericarp thickness. In this experiment, eight recombinant lines provided inbred vigor up to 22%, compared with their respective original hybrids. As for the undesirable characteristic of peduncle size inside the fruit, seven recombinant lines had inbred vigor up to 33%, while all the others showed inbreeding depression that reached 45%.

For tomato fruit quality, reduction of locule number is desirable and a negative estimate of heterosis is valuable. Thus, heterosis breeding can be exploited very well to reduce the number of locules per fruit. These results agree with Kulkarni [76], Prashanth [77], and Duhan et al. [78]. Singh et al. [36] found heterosis of 25% as for locule number. In our study, four recombinant lines possessed inbred vigor up to 14%, while the inbreeding depression of all others reached 34%.

Fruit color is one of the most important fruit quality traits, both in processing and in fresh market tomatoes [79]. Customers prefer tomatoes with intense red color inside the fruit, which indicates for them a high quality. Six recombinant lines of 'Formula', two from 'Elpida' and one from 'Sahara' had higher values than their original hybrid for this trait.

Tomato landraces often have superior fruit quality traits compared to hybrids [80–82]. 'Makedonia' is a dynamic Greek domestic cultivar [45] and it was used as a control for fruit quality evaluations in this study. Although this cultivar had the highest amount of total solids, β-carotene, phenols, and antioxidants, hybrid recombinant lines were better than 'Makedonia' for other fruit quality characteristics, indicating that the recombinant lines had excellent fruit quality.

Pure line cultivars in autogamous crops, such as tomato, carry a very low load of deleterious genes. Continuous selfing and natural or artificial selection in these cultivars has allowed homozygosity to exploit favorable additive gene action, accompanied by

simultaneous gradual removal of deleterious genes. Predominance of pure lines in this group is attributed to the increased amount of the gene product due to additive homoallelic complementation, leading to the so-called inbred vigor [37,83]. As shown from our results, after applying plant breeding methodologies and selecting plants that meet high levels of fruit quality, it is possible to create recombinant lines that show inbred vigor and are equal or even better than the original hybrids. In this study the use of pedigree selection, with honeycomb design, in the first segregating generations of tomato hybrids, was a crucial factor in creating elite inbred recombinant lines. Similar results presented by Fisher [84,85] concluded that there should be new attempts to validate this methodology. The response to selection of the recombinant tomato lines indicates that the characteristics, total soluble content, total solids, β-carotene, carotenoids, lycopene, phenols, antioxidants, pericarp thickness, size of peduncle inside the fruit, locule number, and internal color of fruits in maturity have additive gene action.

## 5. Conclusions

Tomato pure lines with inbred vigor that can achieve or even exceed hybrid vigor in fruit quality characteristics could be created using the appropriate breeding methodology. The honeycomb design applied in this study simultaneously with pedigree selection at the first segregating generations improved selection effectiveness and ultimately produced elite inbred recombinant lines. In tomato, a self-pollinated species with a high load of additive genes, it is possible for inbred vigor to reach and even surpass hybrid vigor. The simultaneous selection of individual plants, both in terms of their high yield and the desired fruit quality characteristics, led to highly productive recombinant lines with integrated quality characteristics. The response to selection for all characteristics evaluated shows the additive gene action for all characteristics measured.

The prevailing tendency in the market is to produce products that meet consumers' high demands regarding their high quality and superior nutritional value. These recombinant lines of tomato presented in this study can fulfill this role as alternatives to hybrid cultivars, and possess high nutritional values to act as a functional-protective food.

**Author Contributions:** Conceptualization, I.D.A. and A.G.M.; data curation, I.D.A., R.T., P.T., I.M., I.N.X. and A.G.M.; formal analysis, I.D.A. and I.M.; investigation, I.D.A., R.T., P.T., I.M., I.N.X. and A.G.M.; methodology, I.D.A., P.T. and A.G.M.; project administration, I.D.A. and A.G.M.; resources, A.G.M.; supervision, I.D.A. and A.G.M.; validation, I.D.A., P.T. and A.G.M..; visualization, I.D.A.; supervision, I.D.A. All authors have read and agreed to the published version of the manuscript.

**Funding:** This research received no external funding.

**Institutional Review Board Statement:** Not applicable.

**Informed Consent Statement:** Not applicable.

**Acknowledgments:** We thank Panagiotis Kalaitzis and Fokion Papathanasiou for their assistance with the initial literature review, their comments and suggestions.

**Conflicts of Interest:** The authors declare no conflict of interest.

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
