# Peer review of "Comparative Evaluation of Tomato Hybrids and Inbred Lines for Fruit Quality Traits"

_agronomy, doi:10.3390/agronomy11030609_

Round 1
Reviewer 1 Report
Dear Editor, Sorry for late response.
Here are my comments to the manuscript: Agronomy-128319, which has been submitted to Agronomy, (Comparative Evaluation of Hybrids and Inbred Lines of Tomato as for Fruit Quality Traits by Avdikos et al.)
The authors, claim, that tomato pure-lines with inbred vigor that can achieve or even exceed hybrid vigor in fruit quality characteristics could be created by using breeding methods mentioned in the manuscript. The authors found the recombinant lines could fulfill the role as alternatives to hybrid varieties. The manuscript contains information that would be of interest to the readers and has high potential bio-economy values. However, authors need to address a hypothesis driven effort and biologically relevant questions to be addressed. What is the author’s hypothesis? More work on the discussion section is needed; see the comments at the end.
General comments;
Do authors have QTL data for these cultivars?
If authors used different soil types and variable field-locations, will it be the same results?
What are the numbers of replicates (biological and technical) used in the study?
Lycopene content was found to be significantly affected by fertilization in Greece. Do authors considered this issue?
Many characters were studied by authors such as; fruit’s pericarp thickness, size of peducle inside the fruit (tables 1, 2, 3, 4 & 5). I strongly recommended a new figure/or figures that summarized all these characters and give better overview for the readers. Much better if authors merge these five tables into 2-3 tables.
The cultivars Iron, Sahara, Formula and Elpida cover 70% of the total tomato cropping area in Greece, this very interesting, but what about other cultivars, which were in pipeline development the 90s.
Line 150, The extraction and the determination of total carotenoids, β-carotene and lycopene content was performed according to Lichtenthaler and Wellburn [42] and D’Souza et al. [43]. A brief and short description of the method is useful and informative for the readers.
How the authors determined the quality of the phenotypic characters (such as colors), did they used HPLC and UHPLC etc.
Line 89, tomato population not to-mato population.
Line 155, The samples not The sam-ples.
<line 308, results not re-sults
Line 191, * Entries with the same letter within a column indicate not significant differences, according to Scott–Knott cluster test. Add the references
Line 213, * Entries with the same letter within a column indicate not significant differences, according to Scott–Knott cluster test. Add the references
Line 240, * Entries with the same letter within a column indicate not significant differences, according to Scott–Knott cluster test. Add the references
Line 283, Breeding efforts in tomato started more than 200 years ago [51]. Is this reference correct!
Line 163, Statistical Analyses, which software program used by authors to perform these statistical analyses?
Discussion
The first two paragraphs should be moved to the introduction section.
Line 361, Macedonia’ is a dynamic Greek domestic cultivar and it was used as a control for the fruit’s quality evaluation. Reference is missing!
Line 315, However, it is an essential source of carotenoids, which are protective against infectious [3,4] and degenerative such as cardiovascular diseases [5] or certain cancers [6,7].
Line 328, Moreover, lycopene is an antioxidant characterized of immuno-stimulatory properties and protects cells against oxidative damage and thereby prevents or reduces the risk of several cancers [62], and cardiovascular disease [63]. Remove it, this is irrelevant to your discussion, these references have nothing to do with your study-discussion. Authors are studying the breeding values of tomato characters NOT tomato as protective agent against infectious to cardiovascular diseases and cancers. Just discuss your results together with other relative studies. There are many more detailed comments that could be made to the discussion. However, I have run out of energy for doing a more thorough review. I leave this to the authors to improve their manuscript on their own. I believe the manuscript contains information that would be of interest to the readers, but need to be improved.
Author Response
We are very grateful for the insightful review of our manuscript. Your remarks and suggestions, which have been incorporated into the text, will significantly increase the scientific value. Below are our point-by-point responses to the concerns raised during the review process.
Do authors have QTL data for these cultivars?
Response: It was a field experiment with evaluation of quality traits. This was the aim of the experiment. So, we have not got molecular data.
If authors used different soil types and variable field-locations, will it be the same results?
Response: The evaluation of genotypes were for qualitative traits that controlled from a small number of genes and affected in a smaller degree from quantitative traits like yield, where the over location experimentation is necessary. The environment of experimentation is a targeted low input environment where tomato is the most important vegetable cultivation.
What are the numbers of replicates (biological and technical) used in the study?
Response: A randomized complete block design (RCBD) experimental design was used, with three replications, with each experimental unit consisting of ten plants (lines 120-122).
Lycopene content was found to be significantly affected by fertilization in Greece. Do authors considered this issue?
Response: The target of this experiment was the low input farming systems. So, we did not make an experiment for fertilization.
Many characters were studied by authors such as; fruit’s pericarp thickness, size of peducle inside the fruit (tables 1, 2, 3, 4 & 5). I strongly recommended a new figure/or figures that summarized all these characters and give better overview for the readers. Much better if authors merge these five tables into 2-3 tables.
Response: In our view, the differences between the genetic materials are presented in a much better way in those Tables.
The cultivars Iron, Sahara, Formula and Elpida cover 70% of the total tomato cropping area in Greece, this very interesting, but what about other cultivars, which were in pipeline development the 90s.
Response: We have data for the best cultivars in Greece and not for the others.
Line 150, The extraction and the determination of total carotenoids, β-carotene and lycopene content was performed according to Lichtenthaler and Wellburn [42] and D’Souza et al. [43]. A brief and short description of the method is useful and informative for the readers.
Response: There is a description. It is a popular method and we have described it in a paragraph.
How the authors determined the quality of the phenotypic characters (such as colors), did they used HPLC and UHPLC etc.
Response: The intensity of the internal color of fruit at maturity (1:weak, 5:strong) was also determined using chromatic maps.
Line 89, tomato population not to-mato population.
Response: It was done. It is an automatic format of Word, that it has been resolved.
Line 155, The samples not The sam-ples.
Response: It was done. It is an automatic format of Word, that it has been resolved.
<line 308, results not re-sults
Response: It was done. It is an automatic format of Word, that it has been resolved.
Line 191, * Entries with the same letter within a column indicate not significant differences, according to Scott–Knott cluster test. Add the references
Response: It has been added in materials and methods section.
Line 213, * Entries with the same letter within a column indicate not significant differences, according to Scott–Knott cluster test. Add the references
Response: It has been added in materials and methods section.
Line 240, * Entries with the same letter within a column indicate not significant differences, according to Scott–Knott cluster test. Add the references
Response: It has been added in materials and methods section.
Line 283, Breeding efforts in tomato started more than 200 years ago [51]. Is this reference correct!
Response: It is correct.
Line 163, Statistical Analyses, which software program used by authors to perform these statistical analyses?
Response: It is referred in section 2.5. (Statistical analysis) line 176
Discussion
The first two paragraphs should be moved to the introduction section.
Response: We have followed your proposal and reduced the size of these paragraphs considering that they are helping the flow of the discussion
Line 361, Macedonia’ is a dynamic Greek domestic cultivar and it was used as a control for the fruit’s quality evaluation. Reference is missing!
Response: Thank you, we have made it.
Line 315, However, it is an essential source of carotenoids, which are protective against infectious [3,4] and degenerative such as cardiovascular diseases [5] or certain cancers [6,7].
Line 328, Moreover, lycopene is an antioxidant characterized of immuno-stimulatory properties and protects cells against oxidative damage and thereby prevents or reduces the risk of several cancers [62], and cardiovascular disease [63]. Remove it, this is irrelevant to your discussion, these references have nothing to do with your study-discussion. Authors are studying the breeding values of tomato characters NOT tomato as protective agent against infectious to cardiovascular diseases and cancers. Just discuss your results together with other relative studies. There are many more detailed comments that could be made to the discussion. However, I have run out of energy for doing a more thorough review. I leave this to the authors to improve their manuscript on their own. I believe the manuscript contains information that would be of interest to the readers, but need to be improved.
Response: In the end of our conclusions we write ‘The prevailing tendency in the market is to produce products that meet consumers' high demands regarding their high quality and superior nutritional value. These recombinant lines of tomato presented in this study can fulfill this role as alternatives to hybrid cultivars, and those that possess high ​​nutritional values to act as a functional-protective food.’ So, the target of the breeding process is to produce tomato cultivars to help with such diseases.
Reviewer 2 Report
This is a very detailed study comparing hybrids and inbred lines of tomato concerning various quality traits. Altogether 11 tomato quality traits were measured. The manuscript is well written, good to understand and follow. The introduction is concise and focuses on the relevant information for the reader. The experimental setup with a randomized complete block design is scientifically sound, the data are well presented. For the results section, I suggest to include an ANOVA table in the beginning, in order to give a quick overview for the reader on the traits which were significantly influenced and which one not. The discussion is good to follow as well, however, in some cases the authors should check for consistency between the results section and the statements in the discussion, please find respective text passages below.
Detailed comments:
Line 2: delete “as”?
Line 17-29: In my opinion, the abstract needs some revision, it should give more details regarding the methods and the experiment itself and also the most important results with some values.
Line 134: delete “of”
Line 163: Please give more details on the statistical analysis, which model was used for ANOVA? Which software was used?
Line 238: Please check the post hoc comparisons (lower case letters) in Table 3 for Phenol content, very small differences in the absolute values, but different letters (a-p) indicating significant difference; even if values are the same
Line 297: I don’t understand this sentence, maybe delete “that”?
Line 316/317: Please rework sentence, difficult to understand
Line 317/318: delete “which”?
Line 333/334: according to the results there was no significant effect on lycopene content? Sentence should be formulated more carefully.
Line 337: “almost all hybrids”: Sorry, I can’t see that in the results (table 3). Only inbred lines of “Formula” reached higher inbred vigour, this statement should be formulated more carefully.
Line 341: “twelve recombinant lines exhibited inbred vigour up to 37 %? I don’t see that in the results (Table 3). For total antioxidants there are only two lines significantly different from the rest of the set. Please check for consistency between results section and the statements in the discussion.
Line 346: see comment above, in most cases this difference is not significant.
References:
- Please check for upper-case and lower-case letters in the titles of your references (see for example line 425 vs line 428, line 445 vs line 448)
- Check for latin plant names in the reference titles, they should be in italics
- Line 571, why here “et al.”?
Author Response
We are very grateful for the insightful review of our manuscript. Your remarks and suggestions, which have been incorporated into the text, will significantly increase the scientific value. Below are our point-by-point responses to the concerns raised during review process.
Line 2: delete “as”?
Response: It was done
Line 17-29: In my opinion, the abstract needs some revision, it should give more details regarding the methods and the experiment itself and also the most important results with some values.
Response: We changed the abstract as you proposed us
Line 134: delete “of”
Response: It was done
Line 163: Please give more details on the statistical analysis, which model was used for ANOVA? Which software was used?
Response: It was done
Line 238: Please check the post hoc comparisons (lower case letters) in Table 3 for Phenol content, very small differences in the absolute values, but different letters (a-p) indicating significant difference; even if values are the same
Response: Phenols is a quality characteristic with low variance, low values and low mean square errors which result in the deferrization of small differences. This is the reason why there is a different letter in each genotype. However, we wrote again the numbers having three decimal places in order to discriminate the same numbers that have different letter.
Line 297: I don’t understand this sentence, maybe delete “that”?
Response: It was done
Line 316/317: Please rework sentence, difficult to understand
Response: It was done
Line 317/318: delete “which”?
Response: It was done
Line 333/334: according to the results there was no significant effect on lycopene content? Sentence should be formulated more carefully.
Response: We do not compare the genetic materials in this case. We write about the breeding term of inbred vigor.
Line 337: “almost all hybrids”: Sorry, I can’t see that in the results (table 3). Only inbred lines of “Formula” reached higher inbred vigour, this statement should be formulated more carefully.
Response: According to Table 3, hybrids Formula, Elpida and Sahara produced recombinant lines (all for Formula, two for Elpida and one for Sahara that possessed inbred vigor. This is what we write in this sentence. That almost all of them had at least one recombinant line with inbred vigor.
Line 341: “twelve recombinant lines exhibited inbred vigour up to 37 %? I don’t see that in the results (Table 3). For total antioxidants there are only two lines significantly different from the rest of the set. Please check for consistency between results section and the statements in the discussion.
Response: We do not speak here for differences of the genetics materials but about the breeding term of inbred vigor.
Line 346: see comment above, in most cases this difference is not significant.
Response: Similarly, we do not speak here for differences of the genetics materials but about the breeding term of inbred vigor.
References:
Please check for upper-case and lower-case letters in the titles of your references (see for example line 425 vs line 428, line 445 vs line 448)
Response: It was done
Check for latin plant names in the reference titles, they should be in italics
Response: It was done
Line 571, why here “et al.”?
Response: It was done
Reviewer 3 Report
The manuscript provides valuable information regarding the value of selected inbreds from hybrid tomato materials for high functional food benefits. Although the data appears sound, this manuscript requires extensive editing to get it into publishable form. So, I have provided a hardcopy with my edits.

Author Response
We are very grateful for the insightful review of our manuscript. All your remarks and suggestions, have been incorporated into the text, and will significantly increase the scientific value. The only point that we did not change, is that in the conclusions where the term ‘inbred vigor’ is very significant and must be presented.
Round 2
Reviewer 1 Report
Dear Editor,
Here are my second comments to the manuscript: Agronomy-128319, which has been submitted to Agronomy, (Comparative Evaluation of Hybrids and Inbred Lines of Tomato as for Fruit Quality Traits by Avdikos et al.)
The authors improved their manuscript and now it is much better than the first version, but I still have two questions;
- What is the author’s hypothesis?
- The authors comment; So, the target of the breeding process is to produce tomato cultivars to help with such diseases (as protective agent against infectious to cardiovascular diseases and cancers ). If this in one of the objective of the study, please add it to the objectives section and proof it!
Author Response
Dear Reviewer
We are very grateful for the insightful review of our manuscript and for your questions you made to us. Your remarks and suggestions, which have been incorporated into the text, will significantly increase the scientific value. Below are our point-by-point responses to the concerns raised during review process.
- What is the author’s hypothesis?
At lines 79-82 is the aim of this study. So, it aims to compare hybrid vigor with inbred vigor in terms of tomato’s fruit quality characteristics. Based on previous research, many of the characteristics associated with tomato fruit quality have additive gene action; so in theory, inbred vigor could be similar to hybrid vigor.
2. As it is written in the Introduction section, many nutraceutical compounds produced by tomato are protective against infectious [3,4] and degenerative human diseases, such as cardiovascular diseases [5] or certain cancers [6,7] and strengthens our immune system [8]. For this reason, there has been recent emphasis on breeding new tomato cultivars with high nutraceutical value. In the bibliography, there are many reports about the nutraceutical value of the tomato compounds and that they are protective against those diseases. It is considered general knowledge so it not necessary to be proved . In tomato breeding, the cultivars with high concentrations of those compounds are likely to be functional-protective food.
Thank you again and we remain at your disposal.